# Vaccination History, Body Mass Index, Age, and Baseline Gene Expression Predict Influenza Vaccination Outcomes

**DOI:** 10.3390/v14112446

**Published:** 2022-11-04

**Authors:** Christian V. Forst, Matthew Chung, Megan Hockman, Lauren Lashua, Emily Adney, Angela Hickey, Michael Carlock, Ted Ross, Elodie Ghedin, David Gresham

**Affiliations:** 1Department of Genetics and Genomic Sciences, Department of Microbiology, Icahn School of Medicine at Mt Sinai, One Gustave L. Levy Place, Box 1498, New York, NY 10029-6574, USA; 2Systems Genomics Section, Laboratory of Parasitic Diseases, NIAID, NIH, Bethesda, MD 20894, USA; 3Center for Genomics and Systems Biology, Department of Biology, New York University, New York, NY 10003, USA; 4Center for Vaccines and Immunology, University of Georgia, Athens, GA 30602, USA; 5Department of Infectious Diseases, University of Georgia, Athens, GA 30602, USA

**Keywords:** influenza, vaccine response, gene expression, random forest

## Abstract

Seasonal influenza is a primary public health burden in the USA and globally. Annual vaccination programs are designed on the basis of circulating influenza viral strains. However, the effectiveness of the seasonal influenza vaccine is highly variable between seasons and among individuals. A number of factors are known to influence vaccination effectiveness including age, sex, and comorbidities. Here, we sought to determine whether whole blood gene expression profiling prior to vaccination is informative about pre-existing immunological status and the immunological response to vaccine. We performed whole transcriptome analysis using RNA sequencing (RNAseq) of whole blood samples obtained prior to vaccination from 275 participants enrolled in an annual influenza vaccine trial. Serological status prior to vaccination and 28 days following vaccination was assessed using the hemagglutination inhibition assay (HAI) to define baseline immune status and the response to vaccination. We find evidence that genes with immunological functions are increased in expression in individuals with higher pre-existing immunity and in those individuals who mount a greater response to vaccination. Using a random forest model, we find that this set of genes can be used to predict vaccine response with a performance similar to a model that incorporates physiological and prior vaccination status alone. A model using both gene expression and physiological factors has the greatest predictive power demonstrating the potential utility of molecular profiling for enhancing prediction of vaccine response. Moreover, expression of genes that are associated with enhanced vaccination response may point to additional biological pathways that contribute to mounting a robust immunological response to the seasonal influenza vaccine.

## 1. Introduction

In non-pandemic years, seasonal influenza generally imposes considerable mortality and morbidity burdens resulting in three to five million severe illnesses per year and 290,000 to 650,000 respiratory deaths worldwide [1]. Public health initiatives to annually administer the seasonal influenza vaccine, targeting predominant circulating influenza strains, are an effective means of mitigating the impact of the virus. However, the effectiveness of the seasonal vaccine—defined as the percent reduction in the frequency of influenza illness among vaccinated individuals compared to the frequency among unvaccinated individuals [2]—shows considerable variation between seasons and among individuals within a season. Between season variation in vaccine effectiveness is primarily attributed to mismatches between the vaccine strain selection and circulating viruses. By contrast, within season variation in vaccine effectiveness is impacted by both viral and host factors, including age, preexisting health conditions such as obesity [3], and prior vaccination and infection history. Understanding the sources of interindividual variation in vaccine effectiveness is critical to the development of robust vaccination strategies.

Host biomolecular factors may be an underlying source of interindividual variation in vaccine effectiveness, specifically antibody repertoires. A number of observational studies provide evidence that pre-existing immunity, due to either prior infection or vaccination, may impact the immune response to vaccination, a phenomenon known as “original antigenic sin” [4,5]). The immune response to infection may be dominated by antibodies to previously encountered hemagglutinin (HA) epitopes, resulting in varying vaccine effectiveness that depends largely on an individual’s age and vaccination history. Additionally, antigenic similarity between consecutive vaccine strains may dampen an individual’s ability to generate antibodies following repeated vaccinations. Antibodies from the initial encounter may mask similar vaccine epitopes in the second encounter, resulting in a diminished response, a phenomenon known as the antigenic distance hypothesis [6,7].

Preexisting immunity and vaccine response can be assessed using the hemagglutination inhibition (HAI) assay, which detects the presence of antibodies that prevent the HA protein of the influenza virus from agglutinating red blood cells [8]. Vaccine response is quantified on the basis of an increase in antibody titer, which can be inferred using the HAI assay for which a 4-fold rise in antibody titers is typically used to define seroconversion [9]. Although not a direct measure of vaccine effectiveness, there is strong evidence that seroconversion is predictive of vaccine effectiveness [10]. However, a caveat to the serological-based assessment of vaccine response using the HAI assay is that it may be inaccurate if most antibodies target the neuraminidase [11]. HAI tests with oseltamivir are used to eliminate such effects [12].

Prior studies have shown that there is considerable variation in the capacity to respond to vaccination [13] with reduced responsiveness associated with biological variables such as sex [14]. Differences in gene expression may be related to, and predictive of, variation in vaccine effectiveness. For example, one large-scale study identified nine genes and three gene modules that show significant expression variation associated with the magnitude of the antibody response [15]. Other studies have identified age-associated differences in gene expression that are associated with differences in vaccine response [16]. These studies indicate that gene expression profiles could be informative about how well an individual will respond to vaccination. However, whether these findings hold in other cohorts and the interactive effects of biological and molecular factors on the predictive value of gene expression in the context of the vaccine response have not previously been assessed.

In this study we sought to determine whether whole blood mRNA expression profiles are informative about pre-vaccination immunity and if they can be used to predict the response to influenza vaccination. We analyzed gene expression prior to vaccination and identified genes that are differentially expressed reflecting differing pre-vaccination immune states. We then tested whether gene expression differences were associated with differences in vaccine response among individuals. We find that a predictive model of vaccine response that uses only the gene expression state of individuals prior to vaccination shows comparable performance to a model that uses only physiological factors (e.g., body mass index (BMI), sex, and age) and knowledge of prior vaccination history. A model that incorporates both gene expression and physiological and demographic factors outperforms either of these models indicating that they contain complementary information. Our study demonstrates that gene expression states prior to vaccination show variation that reflects the preexisting immunological state of individuals. Furthermore, these profiles can enhance prediction of vaccine response when used in combination with physiological information and vaccination history.

## 2. Methods

### 2.1. UGA4 Study

#### 2.1.1. Study Design

Participants were enrolled at the University of Georgia Clinical and Translational Research Unit (Athens, GA, USA) (IRB #3773) from September 2019 to February 2020. All volunteers were enrolled with written, informed consent and excluded if they already received the seasonal influenza vaccine. Other exclusion criteria included acute or chronic conditions that would put the participant at risk for an adverse reaction to the blood draw or the flu vaccine (e.g., Guillain-Barré syndrome or allergies to egg products), or conditions that could skew the analysis (e.g., recent flu symptoms or steroid injections/medications). All participants received a commercially available seasonal influenza vaccine, Fluzone™ (Sanofi Pasteur), which is a split-inactivated vaccine derived from influenza viruses propagated in embryonated chicken eggs. Most participants received a standard dose, quadrivalent vaccine which was formulated with 15 μg HA per strain of A/H1N1 (A/Brisbane/02/2018), A/H3N2 (A/Kansas/14/2017), B/Victoria (B/Colorado/6/2017-like strain), and B/Yamagata (B/Phuket/3073/2013). Some participants 65 years and older chose the high-dose vaccine which was a trivalent composition lacking a B/Yamagata strain, but formulated with 60 μg HA/strain of the others.

#### 2.1.2. HAI Assay

Hemagglutinin inhibition (HAI) assays were performed with serum from each participant pre-vaccination (day 0) and post-vaccination (day 28). Sera was used at a starting concentration of 1:10 following treatment with a receptor-destroying enzyme (RDE) (Denka Seiken) to inactivate non-specific inhibitors. RDE-treated sera (25 μL) were serially diluted in PBS two-fold across 96-well V-bottom microtiter plates to column 11, leaving the last column without sera as a negative control. Similarly treated positive control ferret sera was included on some plates. An equal volume of influenza virus (25 μL), adjusted beforehand via hemagglutination (HA) assay to a concentration of 8 hemagglutination units (HAU) per 50 μL, was added to all wells, plates mixed by agitation, and then incubated at room temperature for 20 min. Next, 0.8% turkey red blood cells (Lampire Biologicals, Pipersville, PA, USA) in PBS were added, plates were mixed by agitation, and then incubated at room temperature for 30 min. The HAI titer was determined by the reciprocal dilution of the last well that contained non-agglutinated RBCs, and a value of 5 was assessed in cases where no HAI was detectable.

### 2.2. RNA Sequencing

#### 2.2.1. Library Preparation and Sequencing

We obtained 275 whole blood samples from the UGA4 study previously described. Each tube contained 2.5 mL of blood that was collected prior to vaccination (day 0) into PAXgene Blood RNA Tubes according to the manufacturer’s protocol (PreAnalytiX, Hombrechtikon, Switzerland). Before freezing, blood was incubated in the tube’s proprietary reagent at room temperature for a minimum of two hours to ensure stabilization of intracellular RNA. Total RNA was isolated from each blood sample in randomized batches using the PAXgene Blood miRNA Kit (QIAGEN, Hilden, Germany) according to the manufacturer’s recommendations and stored at −80 °C in a LoBind twin.tec 96-well PCR plate (Eppendorf, Hamburg, Germany). All samples were quantified with the Qubit BR RNA Assay on a Qubit 2.0 Fluorometer (Thermo Fisher Scientific, Waltham, MA, USA) and their fragment size distribution was measured with an RNA Screentape on a 4200 TapeStation System (Agilent, Santa Clara, CA, USA). Sample yield ranged considerably (min: 0.6 μg, max: 24 μg) but all samples had sufficient input mass for downstream library prep and RIN values were consistently >7.0.

Sequencing libraries were prepared using the NEBNext Poly (A) mRNA Magnetic Isolation Module and NEBNext Ultra II RNA Library Prep Kit for Illumina (NEB, Ipswich, MA, USA) according to the manufacturer’s protocols using either 1 μg of input RNA or, in the case of several low-yield samples, the maximum possible input mass. AMPure XP Beads (Beckman Coulter, Brea, CA, USA) were used in place of the supplied NEBNext Sample Purification Beads. Libraries were prepared in 3 batches with two negative library prep controls in each batch to monitor for reagent or sample-to-sample contamination. Each library was barcoded with NEBNext Multiplex Oligos for Illumina (NEB, Ipswich, MA, USA) in a rotating scheme, which ensured that each pool had a unique set of barcodes. Each library was quantified with the Qubit dsDNA HS Assay on a Qubit 2.0 Fluorometer and fragment size was measured with an HSD1000 Screentape (Agilent, Santa Clara, CA, USA) on a 4200 TapeStation System. All libraries made from experimental samples were of sufficient molarity for pooling and sequencing. Libraries were pooled with equimolar input at either 2 nM or 3 nM into sets of 96 libraries for sequencing on a NovaSeq 6000 with the S1 2 × 150–300 Cycle configuration (Illumina, San Diego, CA, USA) at the Genomics Core Facility (Center for Genomics and Systems Biology, New York University).

#### 2.2.2. Data Processing

Sequence reads were aligned to the human transcriptome using the nf-core/rnaseq nextflow pipeline [17] with default parameters. This pipeline aligned reads to the human genome (GRCh37) using STAR [18] followed by BAM-level quantification with Salmon [19] to generate a feature counts table. All gene expression information and fastq files have been submitted to GEO.

After quantifying human transcripts at the gene level, the count data was processed to keep only one sample per individual with the highest number of gene counts. Additionally, counts from pseudogenes, as annotated by biomaRt in the hsapiens_gene_dataset Ensembl database, were filtered out from any downstream analyses along with the three genes *ENSG00000188536*, *ENSG00000244734*, and *ENSG00000206172*, which correspond to highly expressed hemoglobin genes. TPM values were calculated for all remaining genes using the average gene length of the individual transcripts encoded by their corresponding gene.

#### 2.2.3. Univariate Differential Expression Analyses

Differentially expressed genes as a function of seroconversion score, baseline HAI, age, gender, BMI, race, month vaccinated, and previous vaccination status were separately identified using edgeR v. 3.30.3 [20] with a FDR cutoff of 0.05. For seroconversion score, age, and BMI a fit spline with 1 degree of freedom was used to construct the design matrix while for all other variables, discrete groups were used. Only genes with at least 10 counts-per-million (CPM) in 70% of samples in the smallest group were kept for differential expression analyses. The remaining counts in each sample were normalized by trimmed mean of M-values (TMM) [21] and dispersions were estimated and fit to a quasi-likelihood negative binomial generalized log-linear model (glmQLFit) with robust set to true. UpSet plots [22] used to display intersections of differentially expressed genes were plotted using the R package UpSetR v1.4.0.

GSEA analyses for the sets of genes differentially expressed as a function of seroconversion score and baseline HAI were done using gene lists ranked by a decreasing FC value. GSEA were conducted using the two sets of ranked gene lists individually using the human annotation org.Hs.eg.db v3.11.4, and the R package clusterProfiler v.3.16.1 [23] with the function gseGO with minGSSize = 3, maxGSSize = 800, and a *p*-value cutoff of 0.05. Redundant GO terms were removed using both the simplify function from clusterProfiler with a similarity cutoff of 0.7 (baseline HAI) and 0.4 (seroconversion) and manual curation.

#### 2.2.4. Bivariate Differential Expression Analyses

For seroconversion score and baseline HAI, bivariate differential expression analyses were done using edgeR v. 3.30.3 [20] with a FDR cutoff of 0.05 by constructing a design matrix with one of the two immune terms paired with age, gender, and BMI. A full rank design matrix was unable to be constructed when pairing initial HAI with sex, so those pairings were excluded from downstream analyses. Normalization, dispersion estimation, and linear model fitting were done the same way as for the univariate differential expression analyses. UpSet plots [22] used to display intersections of differentially expressed genes were plotted using the R package UpSetR v1.4.0.

GSEA analyses for both DEGs of seroconversion score and initial HAI as a function of BMI were done using a list of genes ranked by a decreasing FC value. GSEA was conducted using both sets of ranked gene lists individually using the human annotation org.Hs.eg.db v3.11.4, and the R package clusterProfiler v.3.16.1 [23] with the function gseGO with minGSSize = 3, maxGSSize = 800, and *p*-value cutoff of 0.05. Redundant GO terms were simplified using both the simplify function from clusterProfiler with a similarity cutoff of 1 (baseline HAI) and 0.4 (seroconversion) and manual curation.

#### 2.2.5. Multi-Scale Network Analysis

We performed Multiscale Embedded Gene Co-Expression Network Analysis (MEGENA) [1] to identify host modules of highly co-expressed genes in influenza infection. The MEGENA workflow comprises four major steps: (1) Fast Planar Filtered Network construction (FPFNC), (2) Multiscale Clustering Analysis (MCA), (3) Multiscale Hub Analysis (MHA), (4) and Cluster-Trait Association Analysis (CTA). The total relevance of each module to influenza virus infection was calculated by using the Product of Rank method with the combined enrichment of the differentially expressed gene (DEG) signatures as implemented: $G_{j}=\prod_{i}g_{ji}$, where, $g_{ji}$ is the relevance of a consensus j to a signature i; and $g_{ji}$ is defined as $\left(\max_{j}\left(r_{ji}\right)+1-r_{ji}\right)/\sum_{j}r_{ji}$, where $r_{ji}$ is the ranking order of the significance level of the overlap between module *j* and the signature. The correlation between modules and traits was performed using Spearman’s correlation.

### 2.3. Identification of Enriched Pathways and Key Regulators in Transcriptome Modules

The biological functions of identified modules in this study were assessed by enrichment analysis for established pathways and pathway (gene) signatures, including the gene ontology (GO) [24] biological processes (BP) category and MSigDB [25] canonical pathways (C2.CP) [26]. Enrichment analysis was performed using Fisher’s Exact Test (FET; inhouse and the hypergeometric test from the Category R-package).

The analysis to identify key regulators takes as input a set of genes (*G*) and a co-expression network. The objective is to identify the key regulators for the gene sets with respect to the given network. This approach first generates a subnetwork *NG*, defined as the set of nodes in *N* that are no more than h layers away from the nodes in *G*, and then searches the *h*-layer neighborhood (*h* = 1, …, *H*) for each gene in *NG* (*HLN_g,h_*) for the optimal *h*^*^, such that
$$ES_{h}{}^{*}=max\left(ES_{h,g}\right)\;\forall g\in N_{g},\;h\in \left\{1,\;\ldots ,\;H\right\}$$
where *ES_h,g_* is the computed enrichment statistic for *HLN_g,h_*. A node becomes a candidate driver if its HLN is significantly enriched for the nodes in *G*. Candidate drivers without any parent node (i.e., root nodes in directed networks) are designated as global drivers and the rest are local drivers. To identify the system specific key regulators in their coexpression networks, the corresponding SRGs have been used as gene set *G*.

### 2.4. Random Forest Models

Random forest models to predict seroconversion category (low: seroconversion score < 2; high: seroconversion score ≥ 2) were done using the R package randomForest v4.6-14 using a combination of (1) clinical variables including initial HAI, BMI, age, race, sex prevaccination status, and month vaccinated, (2) the 741 genes identified to be differentially expressed as a function of seroconversion, and (3) one of the first five principal components of the identified expression modules. A total of 10 iterations for each input data set were used to generate random forest models. Samples were first split such that 75% of samples were used in the training set with the remaining 25% being used as a test set. The tuneRF function was used, with the options ntreeTry = 500, stepFactor = 1.5, improve = 0.01, using a start mtry value equal to the square root of the number of features. The optimal mtry value was identified as the value at which the out-of-bag error is minimized and stabilized. From this, a random forest model was generated using the optimal mtry value with 500 trees grown. The average out of bag (OOB) error rate and area under the curve (AUC) for the plotted ROC curves of the 10 random forest iterations for a given dataset were used to assess the efficacy of each model in predicting seroconversion category. Cross validation using the test dataset was done using a start mtry value equal to the square root of the number of features and a cv.fold value of 100. The average importance values for each set of features for each random forest model was identified by calculating mean decrease accuracy and the mean decrease in Gini index.

## 3. Results

To investigate the utility of global gene expression profiling in predicting the response to seasonal influenza vaccination, we studied a cohort of participants recruited by the University of Georgia (UGA). The study included a total of 275 participants from a cohort herein referred to as UGA4 that was conducted during the 2019–2020 influenza season. Volunteers provided a blood sample for baseline serology prior to vaccination (Day 0). They were then given a commercially available, seasonal influenza split inactivated virus vaccine (Fluzone, Sanofi Pasteur), and subsequently provided a blood sample 28 days after vaccination to assess response serology. The vaccine was formulated with influenza A strains for H1N1 (A/Brisbane/2/2018) and H3N2 (A/Kansas/14/2017), in addition to influenza B strains for Victoria (B/Colorado/6/2017) and Yamagata (B/Phuket/3073/2013). Standard dose (SD) quadrivalent vaccines (0.5 mL dose with 15 µg HA/strain or 60 µg HA total) were administered to most participants (*n* = 214); however, those participants aged 65 years and older were given the option of a high-dose (HD) formulation (trivalent 0.5 mL dose with 60 µg HA/strain or 180 µg HA total; *n* = 61) in which the Yamagata component was excluded.

The 275 study participants included adult males (*n* = 102) and females (*n* = 173), ranging in age from 18–85 years (mean ± one standard deviation age = 50.3 ± 17.2 years). Self-declared race was recorded: study participants predominantly self-identified as white (*n* = 220), with a small number of black (*n* = 27), hispanic (*n* = 17), asian (*n*= 6), and Native American (*n* = 2) participants. Body mass index (BMI) of participants ranged from 17.43 to 59.1 with 199 of the 275 study participants exceeding a BMI of 25 (Table 1, Appendix A) which is classified as overweight using standard CDC definitions [27]. Participants were assessed for risk factors including smoking (*n* = 94 previous or current smokers) and comorbidities of which hypertension was the most common (*n* = 49). Overall, health metrics are consistent with enrichment for high risk factors among study participants.

Whole blood samples were acquired from participants prior to and following vaccination (Figure 1A). To assess immunogenicity to the influenza HA protein, a hemagglutination inhibition (HAI) assay was performed using two-fold serial dilutions of serum from each participant and turkey red blood cells (RBCs) (Section 2). A numerical HAI value was assigned based on the reciprocal of the final serum dilution at which RBC agglutination inhibition was still observed. For example, if an individual’s serum sample inhibited viral-mediated agglutination of RBCs at a dilution of 1:40, but did not inhibit agglutination at a dilution of 1:80, an HAI score of 40 was assigned. The HAI assay was performed with serum diluted 1:5 using a two-fold dilution series of serum (i.e., 1:10, 1:20, 1:40, …). Thus, HAI values vary from 5 to 1280 with an HAI of 5 corresponding to no detectable antibody titer and higher values reflecting higher antibody titers.

To test the specificity of immunogenicity, HAI assays were performed with each of the four vaccine strains prior to vaccination (day 0, or “baseline HAI”) and four weeks post-vaccination (day 28, or “response HAI”) for all participants (Section 2). However, because the Yamagata strain was not included in the HD formulation the day 28 HAI value was excluded in the serology metrics calculated for those individuals who received the HD vaccine. To account for the different number of strains used in the two formulations we computed average baseline and response serology metrics.

The distribution of baseline HAI values for the four different strains varies between individuals and strains (Appendix A) reflecting significant variation in strain-specific pre-existing immunity among individuals. We observed an increase in the distribution of HAI values for all four strains 28 days after vaccine administration (Appendix A). High response HAI values for each of the four strains reflect both an immunogenic response to the administered vaccine and pre-existing immunity. In general, baseline and response HAI values for each strain show positive correlation (Appendix A). However, individuals with low baseline HAI values show extensive variation in response HAI.

To quantify the response to vaccination, a seroconversion score was computed for each strain by taking the ratio of the response HAI to the baseline HAI. Seroconversion scores were computed separately for each of the four influenza strains and expressed as log_2_ value (i.e., a seroconversion score of 1 corresponds to a twofold increase in HAI value). We find that there is a negative correlation (ρ ≤ −0.2) between baseline HAI and seroconversion for each of the four strains (Appendix A) as high seroconversion scores are generally only observed in those participants with low baseline HAI values. Seroconversion with respect to the Yamagata strain is significantly higher in those individuals who received the SD formulation compared with those receiving the HD formulation, which lacks the Yamagata component, consistent with a specific response to vaccine components (Appendix A). Moreover, with a single exception, none of the participants receiving the HD vaccine had a seroconversion score greater than 2 (i.e., a fourfold increase in HAI) for the Yamagata strain supporting the use of a fourfold increase as indicative of a specific antibody response.

To define an average seroconversion score that incorporates information for each strain we computed the average untransformed seroconversion score for the four (SD) or three (HD) strains and expressed the average as a log_2_ value (adapted from [28]). In general, we treated seroconversion scores as continuous values; however, to increase the statistical power of some analyses, we defined a categorized seroconversion score. Therefore, for these analyses we categorized average seroconversion as “low” (<2, which corresponds to less than a fourfold average change in HAI) or “high” (≥2, which corresponds to a fourfold or higher average change in HAI), consistent with the US Food and Drug Administration Guidance [29]

Average baseline HAI values, incorporating information from all strains, are negatively correlated with average seroconversion scores (ρ = −0.34, *p*-value = 1.12 × 10^−8^; Figure 1B). Among our study participants we find a significant positive relationship between BMI and seroconversion (ρ = 0.21, *p*-value = 0.0005) (Figure 1C) and a slight negative correlation between age and seroconversion (ρ = −0.13, *p*-value = 0.02; Figure 1D). Seroconversion does not differ significantly between males and females (F statistic = 3.435, *p*-value = 0.649).

The majority of study participants received at least one vaccine in the three years prior to the study. However, a small number of participants (*n* = 34) reported no prior vaccination. We find that these individuals have significantly higher seroconversion scores (*t*-statistic = −61.8, *p*-value = 2.2 × 10^−16^) (Figure 1E) consistent with immunologically naive individuals mounting an enhanced response to vaccination.

### 3.1. Differential Gene Expression Analysis

We performed whole transcriptome analysis of whole blood samples for all study participants using RNA sequencing (RNAseq) and processed sequencing data using standard bioinformatic pipelines (Section 2) to generate a table of gene counts (Appendix A). We obtained a median of 15 million reads per sample (Appendix A). Within samples, the distribution of counts per gene is highly skewed as just three genes, encoding hemoglobin, account for almost 50% of the total transcript counts in each sample (Appendix A). Therefore, these three genes were excluded from downstream analyses (Section 2). To identify major sources of variation in the data we performed dimensionality reduction analysis. More than 50% of the variance in the data is captured by the first three principal components (Appendix A). However, visual inspection failed to identify covariates that were non-randomly distributed across these principal components.

We sought to identify genes that show evidence for differences in expression as a function of study participant physiological and serological metrics (Table 1). Therefore, we performed differential gene expression analysis to identify genes with statistically significant differences in expression (methods). Variables that had highly unbalanced categories (i.e., prior vaccination history, and self-reported race) were not considered. We performed differential gene expression analyses using either discrete groups (e.g., for sex) or linear models for continuous variables (e.g., age, BMI, HAI, and seroconversion). We assessed differential gene expression as a function of (1) log_2_ transformed average baseline HAI, to identify genes that are differentially expressed with variation in pre-existing immunity, and (2) log_2_ transformed average seroconversion, to identify genes whose expression prior to vaccination are predictive of the vaccine response.

When considering the pre-existing immune status of individuals we find only 45 genes that are differentially expressed as a function of baseline HAI (Figure 2A, Appendix A). 9 of these genes were uniquely differentially expressed as a function of baseline HAI, while the other 36 were also differentially expressed as a function of different physiological factors. Among the genes that are significantly increased in expression with higher baseline HAI levels are immunoglobulin genes, including *IGLV3-25* (Figure 2B), whereas some genes, such as *LDOC1*, show significant negative expression relationships with baseline HAI (Figure 2C). Another immunoglobulin gene, *IGLV4-69* (logFC = 4.52, FDR = 7.91 × 10^−8^), is also increased in expression with baseline HAI. Other immunoglobulin genes are also among the 45 differentially expressed genes, and include *IVLG8-61* (logFC = 3.88, FDR = 1.89 × 10^−5^), *IGLV4-60* (logFC = 3.32, FDR = 1.59 × 10^−3^), *IGLV3*-*10* (logFC = 2.79, FDR = 4.63 × 10^−3^), and *IGLV3-1* (logFC = 1.71, FDR = 3.1 × 10^−2^) (Appendix A). Among the genes that show the greatest negative relationship to baseline HAI are non-immunoglobulin genes that encode tryptase gamma 1 (*TPSG1,* logFC = −5.82, FDR = 7.91 × 10^−8^*)* and fibromodulin (*FMOD,* logFC = −5.58, FDR = 1.63 × 10^−7^*).* Immunoglobulins are expected to be associated with pre-existing immunity and their increased expression may reflect greater plasmablast cell populations in whole blood from individuals with higher baseline HAI. By contrast, the reduced expression of *TPSG1* with increasing baseline HAI may reflect the role of mast cell protease tryptase gamma 1 in mediation of IL-13/IL-4R/STAT6-dependent proinflammatory pathways via T-cell induction [30]. Quenching of proinflammatory pathways may be required for maintaining a high level of immune response as lower levels of TPSG1 indicate a lower risk of COVID-19 hospitalization [31]. *FMOD* encodes a member of the small interstitial proteoglycans which may play a role in the assembly of the extracellular matrix as well as in regulating TGFβ activity.

When considering differential expression as a function of seroconversion, we identified a set of 84 genes that are uniquely differentially expressed (i.e., show no significant gene expression differences with other participant metrics). A larger set of 741 genes are differentially expressed as a function of seroconversion, but also show significant association with at least one other factor (Figure 2A; note that smaller subsets with fewer than 9 shared genes are omitted to improve visualization quality, Appendix A). These genes exhibit both increased (Figure 2D) and decreased (Figure 2E) expression with increasing seroconversion. The most significantly differentially expressed genes among this gene set include an uncharacterized transcript *ENSG00000273956* (FDR = 7.52 × 10^−22^), and immunoglobulin genes *IGLV4-69* (FDR = 6.07 × 10^−11^) and *IGLV8-61* (FDR = 5.43 × 10^−7^) (Appendix A). Interestingly, *IGLV4-69* has been identified as a component of a monoclonal antibody in a prior vaccination cohort and possesses a broad spectrum binding affinity against a broad range of H3N2 strains, including against the A/Hong Kong/4801/2014 (HK14) vaccine strain used in the 2019–2020 season [32].

To determine the functions of genes that are differentially expressed depending on baseline HAI and seroconversion score, we performed gene set enrichment analysis (GSEA) of the set of significant genes, (Figure 2F). Gene functions that are related to increased baseline HAI include those involved with homeostatic processes. Gene functions that are positively associated with increased seroconversion include immunoglobulin production, and chromatin organization. Conversely, we find gene functions related to growth factor beta responses and neurogenesis to be decreased in expression in individuals with high seroconversion scores. (Figure 2F).

### 3.2. Interactive Effects on Differential Gene Expression

Genes that are differentially expressed as a function of immune status may show differing behaviors depending on other physiological factors. To test for interactive effects between immunological metrics and factors that vary among study participants such as age, BMI, and sex, we used bivariate linear models for both baseline HAI and seroconversion score. We found a number of genes with interactive effects between baseline HAI and sex (1621), age (911), and BMI (111) (Figure 3A). In our bivariate analysis with seroconversion and other covariates, we found the greatest number of interactive effects between seroconversion and age (1834 genes) (Figure 3B). We also found a set of 1048 genes that show significant interactive effects between seroconversion and BMI, and 840 genes that display significant interaction effects between seroconversion and sex (Figure 3B).

We were particularly interested in the interaction between BMI and either baseline HAI or seroconversion as BMI has been identified as one of the major factors associated with increased influenza risk [3]. The differential expression of 111 genes with baseline HAI is influenced by BMI (Figure 3B and Appendix A). For example, *KRT79* increases in expression with higher baseline HAI in overweight individuals, but decreases in individuals with normal BMI (Figure 3C). A similar effect of BMI on the relationship between gene expression and baseline HAI is observed for *IGLV4-60* (Figure 3D). The differential expression of 1048 genes with seroconversion is influenced by BMI (Figure 3B and Appendix A). For example, *PCSK1N* increases in expression with seroconversion in overweight individuals, but not individuals with normal BMI (Figure 3E). Conversely, *DAAM2* decreases in expression with seroconversion in overweight individuals but not in individuals with normal BMI (Figure 3F).

We investigated enrichment of gene sets that show significant interaction between BMI and either baseline HAI or seroconversion. Genes that show interactive effects with BMI and baseline HAI are enriched for immunoglobulin production and immune responses (Figure 3G upper panel). Genes that show interactive effects between BMI and seroconversion are positively enriched for diverse functions including metabolism, immunoglobulins and the adaptive immune response, and T cell receptor functions (Figure 3G lower panel).

### 3.3. Identification of Gene Expression Modules Related to Serological Outcome

We sought to identify molecular processes that show coherent gene expression behavior among individuals. Therefore, we defined sets of genes that show correlated co-expression patterns across individuals using MEGENA (Section 2). The MEGENA method identifies modules of correlated gene expression across samples and defines a hierarchical relationship between modules defined by different levels of stringency. Using the 275 transcriptomes, we identified 184 hierarchically ordered expression modules containing between 5 to 2450 genes.

We employed different measures based on correlation, differential gene expression enrichment, and a combination of both, to quantify the association between modules and physiological and serological metrics. We use the product of rank method (Section 2) to compute an aggregated rank from individually ranked modules. The top ten highest ranked modules after enrichment of modules for baseline HAI, response HAI and seroconversion differentially expressed gene (DEG) signatures, are related to the adaptive immune response and immunoglobulin complexes (Figure 4). These modules are strongly enriched for genes that are differentially expressed as a function of response HAI, but show only marginal enrichment for genes that are differentially expressed as a function of baseline HAI and seroconversion (Figure 4A). However, we find significant positive correlation between these modules and all three serological scores (Figure 4B). The hierarchical organization of five of the top ten modules is M11 → M64 → M175 and M23 → M116, where M11 is the parent module that includes each subset of smaller modules (Figure 4C). Module M11 contains many immunoglobulin genes that show increased expression in subjects with high seroconversion (Figure 4D). The most highly connected, or hub genes, of M11 are the variable chain immunoglobulin genes *IGHV5-51*, *IGKV1-5*, *IGKV3-11*, *IGKV3-15*, *IGKV3-20*, *IGKV4-1*, and *IGLV1-40*. It is noteworthy that a proteomic analysis of influenza haemagglutinin-specific antibodies following H1N1pdm09 influenza A vaccination identified *IGKV3-20* as the dominant light chain variant [33]. Modules defined by the M11 module family are enriched for functions related to the adaptive immune response (Figure 4E). Interestingly, these modules are also strongly negatively correlated with age (Figure 4B). This points to a potential indirect effect of age on serological outcome manifested in the expression state of these immunoglobulin-enriched modules that may underlie the observed negative relationship between age and seroconversion (Figure 1D).

Additional highly ranked modules are suggestive of other relationships between gene expression and serology. For example, module M37 is enriched for inflammatory response and is negatively correlated with seroconversion (Figure 4C) and positively correlated with baseline HAI. The module family M23 → M116, is enriched for mRNA 5′-splice site recognition, lyase activity regulation and extracellular matrix function and is positively correlated with baseline HAI. The most significant module relationships are found with age and BMI (Appendix A). The module with the most significant correlation with age is M192 (ρ = −0.63, FDR = 4.58 × 10^−30^), which contains 15 genes, including the type I transmembrane glycoprotein *CD248*, Rho guanine nucleotide exchange factor 4 (*ARHGEF4*), Wnt signaling factor regenerating family member 4 (REG4). M192 also has fibroblast growth factor 1 (*FGFR1*) functionality (*PTK7*, *CD248*, *PHGDH*, *FBLN2*; FDR = 0.006).

With respect to age association, most modules are enriched for age-specific differential gene expression (Appendix A) and all are strongly correlated with age (Appendix A). Interestingly, the modules with the most significant enrichment for age-specific DEGs are the same as the highest ranked modules based on the serological response (Figure 4). The module M11 → M64 → M175, which is enriched for immunoglobulin functions, is negatively correlated with age and positively correlated with baseline HAI, response HAI, and seroconversion. Although M192 is enriched for GO functions related to nervous system development, it involves hub-gene CD248 related to the promotion of CCL17 expression in pro-fibrotic macrophages (Appendix A). Another example of an age-related DEG enriched module is M311. This module is positively correlated with age and does not show any significant correlation with the serological responses. M311 is enriched for molecule transducer and signaling receptor activity with serine protease 23 (*PRSS23*) as a key regulator (Appendix A).

With respect to BMI, the highest ranked module is M10 with basophil/mast cell regulation functions. M10 and related module M62 are weakly positively correlated with Baseline HAI values (Appendix A). Key regulators histidine decarboxylase (*HDC*) and IgE-receptor β-subunit *MS4A2* modulate immune response (Appendix A). The IgE-receptor is expressed on basophils and mast cells, required for a robust vaccination response [34]. A second set of key regulators involve A-kinase anchoring protein 12 (*AKAP12*) and GATA binding protein *GATA2*. Both genes are responsible for cell proliferation and migration. A second example of a module enriched for BMI DEGs is M13 (Appendix A). This module is positively correlated with both age and BMI but not with serological responses. M13 is enriched for immune effector process functions involving defensins including the key regulator, CEA cell adhesion molecule 8 (*CEACAM8*), a cell-adhesion protein in neutrophils, and defensin *DEFA4*, which is involved in host defense.

### 3.4. Prediction of Vaccine Response

We tested the predictive value of physiological information and gene expression with respect to seroconversion. For this purpose, we generated random forest models using different combinations of three distinct classes of features: (1) physiological metrics, (2) expression of genes that are differentially expressed as a function of seroconversion, and (3) co-expression modules. For our set of differentially expressed genes, we tested three subsets to account for the possibility of overfitting. We used gene sets consisting of (1) all 741 genes differentially expressed as a function of seroconversion, (2) the top 75% of seroconversion differentially expressed genes sorted by FDR, the same fraction used for our training set, and (3) a set of 84 genes differentially expressed exclusively as a function of seroconversion and no other physiological data. We initially attempted to predict quantitative seroconversion scores, but determined that all models performed poorly. Therefore, we classified individual seroconversion scores as high (average seroconversion ≥ 2, *n* = 124) or low (average seroconversion < 2, *n* = 151) for the purpose of prediction.

When used to predict seroconversion, we find that a random forest model generated using only physiological metrics results in the lowest out-of-bag (OOB) error rate. The random forest models generated using only sets of differentially expressed genes yield the highest OOB error rate whereas the combined data results in an intermediate error (Figure 5A). We then evaluated the accuracy of each model’s average predictions by using a receiver operating characteristic (ROC) curve and calculating the resulting area under the curve (AUC). The random forest model generated using a combination of the physiological data and the expression data of the set of 84 genes exclusively differentially expressed as a function of seroconversion yielded the highest average AUC of 0.73 while all other models have AUCs less than 0.70 (Figure 5B).

To confirm that our results are reproducible, we repeated our OOB and AUC tests varying the number of initial trees from 50 to 5000. We obtained similar results for each test in that the model generated using only physiological data had the lowest OOB error while the model generated using physiological data combined with the set of 84 DE genes had the best AUC regardless of the number of initial trees (Appendix A). Of our data sets that only use expression data, the smallest set of 84 genes that were only differentially expressed as a function of seroconversion performs the best. Similarly, when combining expression data with physiological data for our models, using the set of 84 genes results in the best performance.

To identify the specific instances for which our random forest models fail, we generated confusion matrices using the average predictions of each of three of the datasets consisting of (1) only physiological data, (2) expression data of the 84 exclusively differentially expressed genes, and (3) the two data sets combined (Figure 5C). All three random forest models have difficulty correctly categorizing high seroconversion, often mistakenly categorizing them as low. However, when assessed using the test dataset, the random forest model generated using only physiological data accurately classifies 53.3 of the 93 high seroconversion individuals on average compared to the models generated using only expression data (42.6 of 93 correctly assigned) and clinical data paired with expression data (47.1 of 93 correctly assigned), respectively.

We performed cross validation analyses for each of the three random forest models to assess the reproducibility of the model on unseen data as well as to identify the optimal number of features for each model. We find that for models that contain physiological data the error is lowest when less than 5 features are used (Figure 5D). By comparison, the random forest generated using only gene expression data requires almost all features to obtain a similar predictive error. This suggests that physiological information is more informative about serological response than baseline gene expression. For each random forest model, we calculated and ranked the average importance value for each feature (Figure 6). When using a model with only physiological metrics to predict seroconversion, prior vaccination status is the most important variable (Figure 6A), but BMI, baseline HAI, sex and age also contribute to model performance and reproducibility as measured using the GINI coefficient (Figure 6B). When using only the set of 84 differentially expressed genes we find that removal of individual genes has incremental effects on model performance as measured by the decrease in model accuracy (Figure 6C) and GINI coefficient (Figure 6D) as do models that combine physiological metrics and differentially expressions genes (Figure 6E,F). This is consistent with no single gene having an expression value that is especially information of seroconversion. However, we do find that the non coding RNAs *ENSG00000267009*, *LINC01960*, negative regulator of antiviral response *NRAV* as well as complement C8B, novel protein *ENSG00000255835*, trophinin associated protein *TROAP*, and planar cell polarity protein 2 *VANGL2* consistently make the most significant contribution to seroconversion prediction.

We further investigated whether prediction of seroconversion outcome would improve with the inclusion of the aggregated module data generated using MEGENA. Combination of one of principal components 1–5 of the module data from the top 20 ranked modules with the physiological data and DEGs yields a comparable OOB error rate but a decreased AUC compared to random forest models generated with physiological data with DEGs (Appendix A). An exception is the contribution by the principal component 2 of the top ranked modules, which results in a similar AUC compared to the physiological data plus DEG only models. However, we did not observe any improvement in prediction by including module contributions. Similarly, ranking variables by importance in this set of random forest models consistently indicates previous vaccination status to be the most important variable in predicting seroconversion category (Appendix A). Of the clinical variables, baseline HAI, BMI, and previous vaccination status consistently are in the top 10 variables for this set of random forest models. While no modules appear in the top 10 most important features across multiple random forest models in this set, similar to the models without module inclusion (Figure 6) we do see the genes *C8B*, *ENSG00000267009*, *LINC01960*, *NRAV*, and *TROAP* show up in the top 10 features in multiple random forest models in this set (Appendix A).

When combining physiological features with gene expression, previous vaccination status, BMI, and baseline HAI remain highly predictive along with above mentioned genes and lncRNAs in addition to ENSG00000255835, *NCBP2L*, *NKX3.1*, *PTGER1*, and *VANGL2*. Complement factor *C8B*, *NRAV* and *PTGER1* have immune system relevant functions. *C8B* is the β-chain of the membrane attack complex, which mediates cell lysis, and initiates membrane penetration of the complex. The lncRNA *NRAV* modulates antiviral responses through suppression of interferon-stimulated gene transcription [35]. Prostaglandin E receptor 1 (*PTGER1*), a member of the G protein-coupled receptor family, plays a role in inflammatory response and modulates inflammation by down-regulation of COX-2 [36]. Overall, the main biological functions of these genes are related to immune processes.

## 4. Discussion

The immunological response to the seasonal influenza vaccine exhibits significant heterogeneity among individuals. In this study, we addressed two key questions with respect to the observed variation: (1) to what extent do whole blood gene expression profiles and physiological variables relate to serological status prior to, and following, vaccination, and (ii) are gene expression and physiological metrics predictive of vaccine response.

We first assessed the effect of physiological factors on serological status. As expected, the response to the vaccine is strongly impacted by existing immunity. We quantified existing immunity to each of the four influenza strains included in the vaccine and observed significant variation among individuals. Among those individuals with higher existing immunity we detect a muted increase in immune status following vaccination. Individuals who had not been vaccinated in the prior three years show a much greater response than those who received at least one influenza vaccine in the prior three years. Surprisingly, we detect a trend of higher BMI associated with increased vaccine response. This could reflect either systematically decreased pre-existing immunity, or a greater degree of immunological priming, in higher BMI individuals.

Gene expression profiling revealed a large number of genes that are differentially expressed with physiological factors including BMI (3814 genes), sex (2977 genes), and age (2977 genes). By contrast, only 45 genes are differentially expressed as a function of baseline HAI. Surprisingly, a much larger number of genes are differentially expressed with seroconversion (741) although only 11% (84) of these are not associated with one of the physiological factors. This discrepancy suggests that whole blood gene expression profiles obtained prior to vaccination are more informative of the response to vaccination than the pre-existing immunological state of an individual.

Among genes that are increased in expression with increasing seroconversion are several immunoglobulin genes. As gene expression is assayed prior to vaccination this may indicate that those individuals who will mount the greatest serological response to the vaccine are predisposed to do so as a result of increased pre-existing immunological activity [37,38]. It is possible that these expression values may be influenced by imprecise mapping of RNAseq reads to variable gene segments of immunoglobulins. However, we believe that this is unlikely as visual inspection of aligned reads for a representative example (*IGLV4-69*) indicates high quality mapping and BLAST searches of *IGLV4-69* mapped reads against the reference genome identify this feature as the most likely source. Our findings suggest the interesting possibility that enhanced vaccine response may result from either increased plasmablast subpopulations or enhanced expression of specific immunoglobulins. Testing these hypotheses requires additional analyses, possibly through variable sequence reconstruction using RNAseq reads [39,40], and cell profiling of participants.

Using a network-based approach we identified co-expression modules that were significantly associated with serological status. As with our gene-level analysis, we identified modules involving immunoglobulins. Module M11 with hub genes *IGHV5-51*, *IGKV1-5*, *IGKV3-11*, *IGKV3-15*, *IGKV3-20*, *IGKV4-1*, and *IGLV1-40*, together with modules M64 and M175 are positively correlated with baseline HAI, response HAI and seroconversion. These modules are functionally enriched for antigen binding, immunoglobulin complex, and activation of adaptive immune response. We further observed negative correlations between these immunoglobulin modules and age, which may reflect reduced immunity with age.

Central to our analysis was the identification and assessment of factors for the prediction of seasonal influenza vaccine response. For this purpose we employed a random forest approach for both feature selection and modeling. Consistently, the most important physiological predictive features are baseline HAI, BMI, and prior vaccination status. When combining physiological features with gene expression, these three features remain highly predictive along with genes and lncRNAs, such as *C8B*, *ENSG00000267009*, *ENSG00000255835*, *LINC01960*, *NCBP2L*, *NKX3.1*, *NRAV*, *PTGER1*, *TROAP*, and *VANGL2*. Complement factor C8B, NRAV, and PTGER1 have immune system relevant functions. *C8B* is the β-chain of the membrane attack complex, which mediates cell lysis, and initiates membrane penetration of the complex. Complement activation is associated with vaccine efficacy, although the exact mechanisms are still unclear [41]. A recent study suggested active modulation of glycosylation of serum glycoproteins by the immune system to establish effective post vaccine protection via complement activation [42]. The lncRNA *NRAV* modulates antiviral responses through suppression of interferon-stimulated gene transcription [35]. Prostaglandin E receptor 1 (*PTGER1*), a member of the G protein-coupled receptor family, plays a role in inflammatory response and modulates inflammation by down-regulation of COX-2 [36]. Among the other, not directly immune-system related genes are three with unknown functions, two lncRNA, *ENSG00000267009* and *LINC01960,* and one protein coding gene, *ENSG0000025583* with hypothetical pyrroline-5-carboxylate reductase function. These predictive but unknown molecular factors may require further investigation in their role affecting vaccination outcome. However, despite the predictive value of gene expression we find that response prediction using exclusively physiological information performs better than models that include gene expression, either exclusively or additionally.

Overall, our study further highlights the importance of physiological variables on the vaccine response, and identifies key genes and co-regulatory networks associated with the individual vaccine response. Whereas physiological factors, such as prior vaccination status, age, and BMI are major determinants of vaccination outcome, we demonstrate that an individual’s baseline gene expression also impacts their response to vaccination. This finding has important implications for the prediction of vaccine response, and may point to biological pathways that are critical for mounting an effective immune response to influenza vaccination.

## Figures and Tables

**Figure 1 viruses-14-02446-f001:**
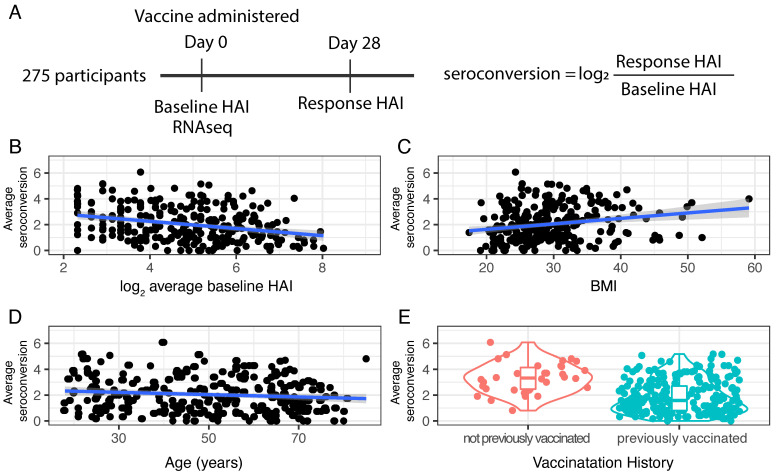
Study design and response to vaccination. (**A**) 275 participants from the UGA4 vaccination study were selected for whole blood gene expression analysis. HAI for four strains was determined prior to vaccination (baseline HAI) and 28 days (response HAI) after vaccination and used to compute an average seroconversion score. (**B**) There is a significant negative relationship between average baseline HAI and average seroconversion (ρ = −0.34, *p*-value = 1.12 × 10^−8^). (**C**) There is a significant positive relationship between BMI and average seroconversion (ρ = 0.21, *p*-value = 0.0005). (**D**) There is a slight negative relationship between age and average seroconversion (ρ = −0.13, *p*-value = 0.02). (**E**) Those individuals who were not previously vaccinated have higher average seroconversion scores than previously vaccinated individuals.

**Figure 2 viruses-14-02446-f002:**
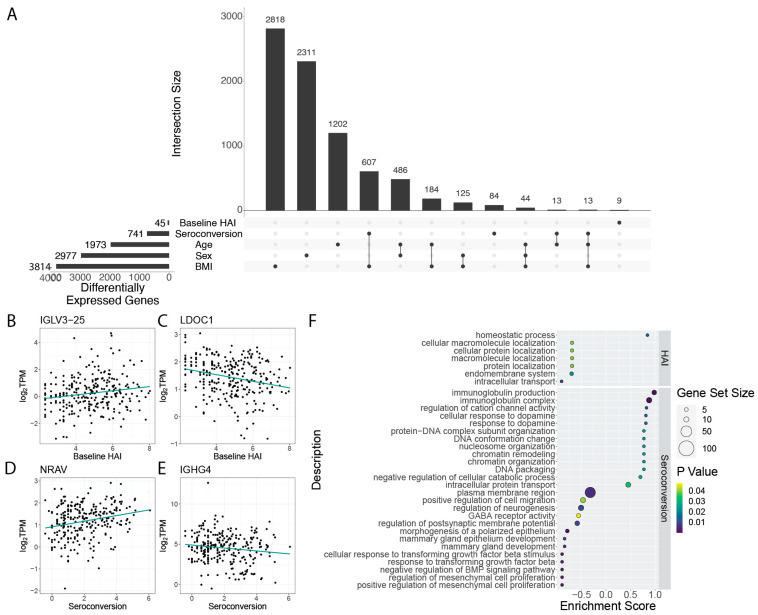
Differential gene expression analysis of pre-existing immunity and vaccine response. (**A**) Genes that are differentially expressed with variation in baseline HAI and seroconversion. An UpSet plot shows the top 12 subsets ranked by size of differentially expressed genes grouped by physiological features. Sets of genes differentially expressed with two or more variables are connected by lines. Representative genes with (**B**) increased and (**C**) decreased expression in participants with higher baseline HAI. Representative genes that are (**D**) increased and (**E**) decreased in participants with higher seroconversion. (**F**) Gene function enrichment using GSEA of the 45 genes differentially expressed with baseline HAI (**top**) and the 741 genes differentially expressed with seroconversion (**bottom**). Note that the histogram in panel (**A**) has been truncated for values less than 9 for visualization purposes.

**Figure 3 viruses-14-02446-f003:**
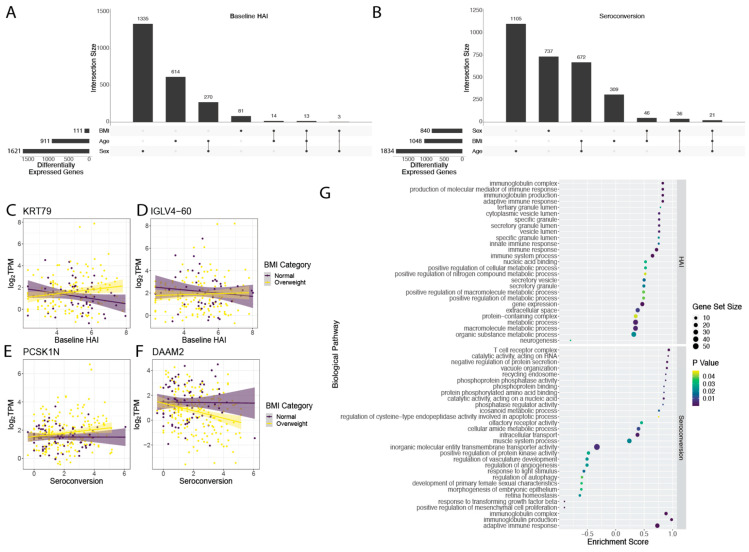
Interactive effects of physiological variables on differential expression with baseline HAI and seroconversion. Sets of genes showing interactive effects with sex, BMI and age on differential gene expression with (**A**) baseline HAI and (**B**) seroconversion. Representative genes in which BMI impacts (**C**) positive and (**D**) negative differential expression with baseline HAI. Representative genes in which BMI impacts (**E**) positive and (**F**) negative differential expression with seroconversion. (**G**) Gene function enrichment using GSEA of 111 genes that show interactive effects with BMI and baseline HAI (**top**) and 1048 genes that show interactive effects with BMI and seroconversion (**bottom**).

**Figure 4 viruses-14-02446-f004:**
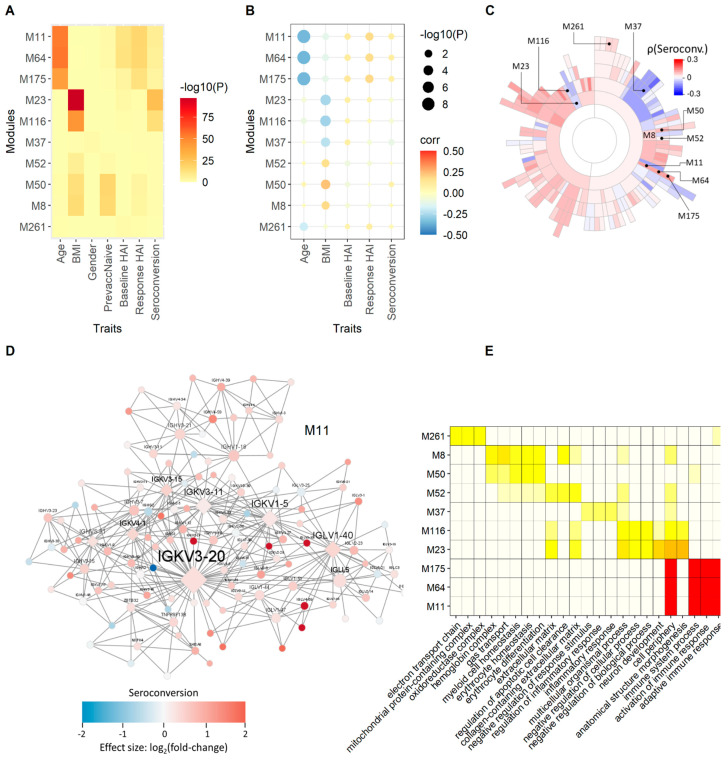
Multi-scale co-expression network analysis. Using MEGENA we defined gene co-expression network modules associated with participant metrics and immune status. The 10 best ranked modules are shown using a combined rank based on the correlations related to serological effects and outcome (average baseline, average seroconversion and average response). (**A**) The module enrichment for DEGs are depicted in this heatmap. (**B**) The dot-plot shows the correlation coefficients and corresponding *p*-value of module/trait correlations. (**C**) A sunburst plot with the hierarchical structure of the module. (**D**) Module M387 enriched for immunoglobulin functionality. Node colors indicate fold change with respect to high versus low average seroconversion, red for increased expression in subjects with high seroconversion compared to subjects with low seroconversion, blue refers to decreased gene expression between these two groups of subjects. (**E**) The functional enrichment of the corresponding modules for GO functions are shown. The hierarchical relationship of the modules shown in (**A**,**B**) are as follows: M8 → M50/M52; M11 → M64 → M175; M23 → M116.

**Figure 5 viruses-14-02446-f005:**
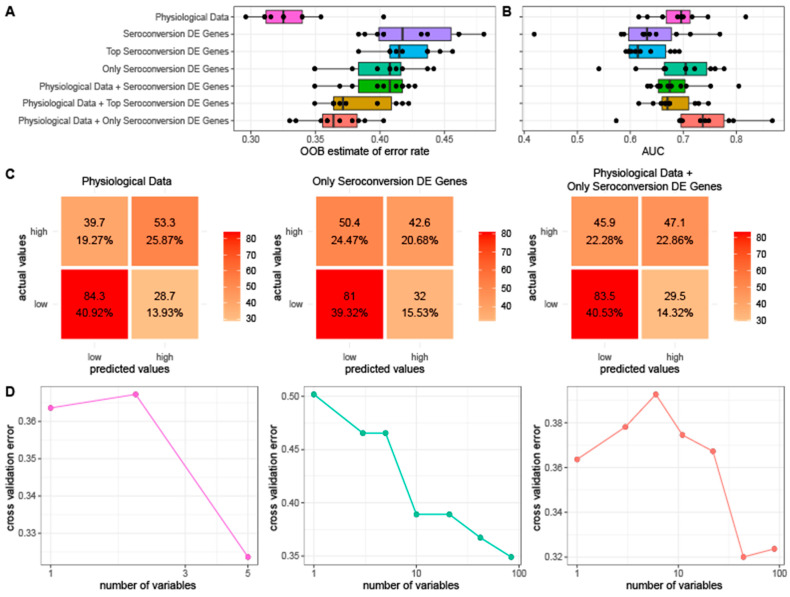
Prediction performance of random forest models generated using physiological and differential gene expression data to predict seroconversion. The average (**A**) out-of-bag (OOB) error rate and (**B**) average area under the curve (AUC) values generated using 10 iterations of random forest models constructed using a combination of physiological data, expression data, and both combined. (**C**) The confusion matrices displaying true and false positives and negatives for the random forest models using physiological data (**left**), expression data from the 84 exclusively differentially expressed genes as a function of seroconversion, (**middle**), and both combined (**right**). (**D**) A cross validation analysis plotting a varying number of variables for the physiological data (**left**), expression data from the 84 exclusively differentially expressed genes as a function of seroconversion (**middle**), and both combined (**right**) random forest models against the resulting cross validation error.

**Figure 6 viruses-14-02446-f006:**
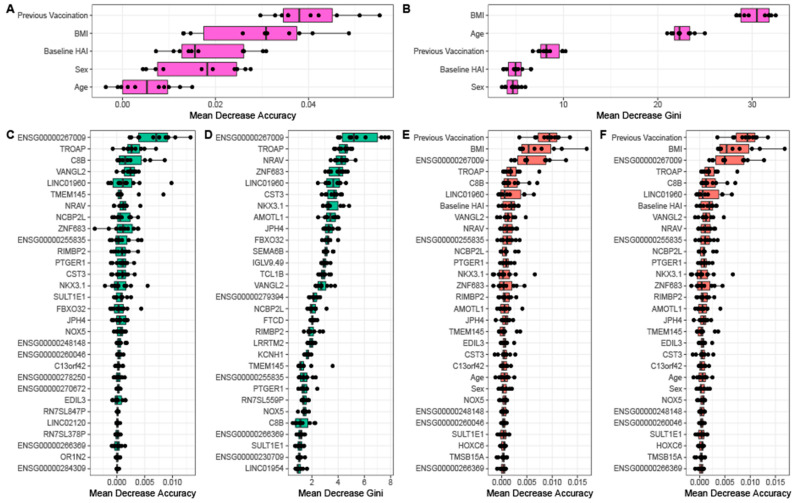
Identification of factors underlying random forest model performance. Variable importance measured through mean decrease accuracy and mean decrease in Gini coefficient for random forest models generated using participant (**A**,**B**) physiological metrics, (**C**,**D**) the 84 DEGs as a function of exclusively seroconversion, and (**E**,**F**) physiological metrics and the 84 DEGs combined.

**Table 1 viruses-14-02446-t001:** Demographic and health metrics of study participants. A total of 275 participants in the UGA4 (2019–2020) split inactivated virus (Fluzone, Sanofi Pasteur) influenza vaccine trial were included in the study.

Metric	Categories	Mean (+/−Standard Deviation)	Range
**Age**		50.3 (± 17.2 years)	18–85 years
**Sex**	Males 102 Female 173		-
**Race/Ethnicity**	White 220 Black 27 Hispanic 17 Asian 6 Native American 2 Multiracial 3	-	-
**BMI**	Overweight (BMI ≥ 25) 199 Healthy (BMI 18.5–25) 76	29.36 (±6.66)	17.43–59.1
**Smoking**	Previously 73 Current 21 Never 179 N/A 2	-	-
**Comorbidities**	Hypertension 43 High cholesterol 23 Depression 17 Type II diabetes 12 Sleep apnea 31	-	-
**Prior vaccination**	Vaccinated 2018–2019 231 Vaccinated 2017–2018 211 Vaccinated 2016–2017 193 Not in prior 3 years 34	-	-

## Data Availability

All computer code to reproduce all figures and analysis in this study is available on Github (https://github.com/GreshamLab/CIVR_HRP_Day0). The data discussed in this publication have been deposited in NCBI’s Gene Expression Omnibus and are accessible through GEO Series accession number GSE207750 (https://www.ncbi.nlm.nih.gov/geo/query/acc.cgi?acc=GSE207750).

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
