# Peer review of "Vaccination History, Body Mass Index, Age, and Baseline Gene Expression Predict Influenza Vaccination Outcomes"

_viruses, 2022, doi:10.3390/v14112446_

Round 1
Reviewer 1 Report
Seasonal influenza vaccination is a practice that helps to reduce mortality and complications from the disease. Studying factors affecting its efficiency is important for ensuring equitable outcomes in medical care and is an important step for personalized healthcare. The authors conduct an important study to uncover genetic factors underlying the variability in the vaccine effects. They performed mRNA studies on serological samples from patients and generated a new valuable dataset. Afterward, the authors performed computational analysis by building a predictive model. The authors also identify relevant molecular processes based on correlated expression patterns across individuals. Overall, the quality of work and text is high.
However, there are some issues with the machine learning part of the study. The performance of the combined model (genetic + physiological information) deteriorates in comparison both to the simple physiological-based model and the genetic-based model. Such results are counterintuitive and require further exploration and suggest that a feature selection procedure can be significantly improved. Here are some suggestions:
- Authors report a decline in predictive power for the random forest (RF) model with the addition of genetic features (Fig. 5B). Indeed, the size of the full gene set is 741, which is much larger than the available number of samples - 275. This situation likely leads to overfitting, which would explain results deteriorating in comparison to the much simpler five-factor-based physiological model. I would ask the author to reduce the gene set for RF model to 206, a size of the training set based on the presented methodology (275 * 0.75). The authors have already performed an initial seroconversion analysis using clinical data and explored overlap in DE genes across seroconversion and other functions. This approach is a useful feature selection step for a predictive model, as it eliminates a large amount of confounding factors, and a reduced set of 84 genes have a high chance of providing additional information on top of physiological features.
- The authors mention the usage of RF as a method for both feature selection and modeling. However, in a scenario when the number of features is much larger than the number of samples, RF model is likely to be overfitted. Even though tuning mtry parameter may help to address this issue, it is not evident that this issue was eliminated in the current study. The default recommendation for mtry parameter is a square root of the number of features. The authors start from 1/3 of the number of features and tune this parameter, but it is not clear if it reaches an optimal value. Also, if the authors would like to stick with RF for feature selection, I would like to direct them to the specialized RF-based schemes for high dimensionality and low-sample size.
- As mentioned in 1, authors can use a reduced 84 gene set for their analysis. The way it was curated is really great for building a predictive model in tandem with five physiological functions. The authors can also explore other feature selection methods, e.g., univariate ones. If some of the selected genes are associated with physiological function, it would be informative to take a look at the additional value they introduce in comparison to the clinical data. This can be also used as an additional criterion for feature selection.
- The effects of the number of trees in RF are not explored. This model may be sufficiently complicated to fit physiological data, but may fail to make full use of genetic information.
Minor comments:
- In the results section on line 247 authors report a larger set of 741 genes that are differentially expressed as a function of seroconversion. However, there are some discrepancies in numbers for an UpSet plot (figure 2A). The total number of differentially expressed genes relevant to the separate function (baseline HAI, age, etc.) does not equal to the sizes of sets presented on the plot. So, for seroconversion, the total amount of DE genes across all subsets is 717 (607 + 84 + 13 + 13). Are there any subsets that are hidden, e.g., subsets spanning across 4 or more functions? It is perfectly valid to do this to improve visualization quality; however, a brief clarification of this discrepancy would be helpful to the reader.
- Supplementary materials (Fig. S17) describe the usage of 1,084 DEGs as a function of seroconversion in the RF model. It is inconsistent with the previously established set of 741 DE genes as a function of seroconversion.
Reviewer 2 Report
General remarks
The manuscript "Predicting seasonal influenza vaccine response using systemic gene expression profiling" by Christian W. Forst et al. is devoted to such a topical topic as studying the main processes of gene expression in individuals depending on the physiological characteristics of the body in response to vaccination with the influenza virus. The authors used modern approaches of bioinformatics methods. Methods and results are described in detail; conclusions are correct. The manuscript is consistent with the purpose and scope of Viruses and may be published after some issues have been resolved.
Special remarks
1-2. In my opinion, the title of the article significantly reduces the significance of the presented results.
41. CDC 2021b
The reference in the bibliography is incorrect.
53. Kobe & Hensley 2017, Chan & Ross 2019
This item is missing from the bibliography.
60 Smith et al. 1999, Skowronski et al. 2016
This item is missing from the bibliography.
66. Katz, Xu, 2011
One of the authors (Hancock) is omitted.
66. Kaushemez, 2012
This item is missing from the bibliography.
68-70. To eliminate the effect of anti-NA antibodies, the HAI test with oseltamivir is used. For example, see Cheng Y., et al. Effect of oseltamivir on the hemagglutination test and the influenza A(H3N2) hemagglutination inhibition test in China. Bing Du Xue Bao. 2017 Jan;33(1):13-18. English, Chinese. PMID: 30702816.
91. BMI should be explained at first mention.
96-98. The statement of the purpose of the study seems to be too simplistic.
136. After treatment of sera with SERUM dilution is 1:10. Therefore, the initial dilution cannot be 1:5.
165. One point in figure S5 is above 2.
170-171. This refers to the data obtained in various laboratories. Other works indicate the high reproducibility of the results. See, for example, Joanna Waldock et al., Assay Harmonization and Use of Biological Standards To Improve the Reproducibility of the Hemagglutination Inhibition Assay: a FLUCOP Collaborative Study. 2021 July-August; 6(4): e00567-21. Published online 2021 Jul 28. doi: 10.1128/mSphere.00567-21.
376. Key regulators
It should be removed.
554-557. Why information on the physiological characteristics of the person is absent if this vaccinated person is known?
552-557. The authors presented sound baseline work on the study of individual response to frequent vaccination with a highly variable antigen. They got interesting and important basic data. I recommend the authors to deepen the main conclusion of the study.
Figure S1. Figure S2. The x-axis indicates log2 but not the HAI value.
Round 2
Reviewer 1 Report
I thank the authors for addressing all issues.